# Improving Mechanical and Tribological Behaviors of GLC Films on NBR under Water Lubrication by Doping Ti and N

**Zhen Zhou [1,2], Yanfeng Han [1,2,\*] and Jin Qian [3]**

1    College of Mechanical and Vehicle Engineering, Chongqing University, Chongqing 400044, China; 20160701005z@cqu.edu.cn
2    State Key Laboratory of Mechanical Transmission, Chongqing University, Chongqing 400044, China
3    College of Optoelectronic Engineering, Chongqing University, Chongqing 400044, China; k7878_@126.com
\*    Correspondence: yfhan@cqu.edu.cn

**Abstract:** Water lubrication has been widely used in marine equipment, where rubber bearings and seals suffer intense friction and severe wear under mixed and boundary conditions. It has good research prospects and practical value to study the composite of amorphous carbon on water lubrication rubber to improve lubrication and reduce wear. In this work, modified graphite-like carbon films incorporated with titanium and nitrogen ((Ti:N)-GLC) were integrated on nitrile butadiene rubber (NBR) with multi-target magnetron sputtering. Direct current (DC) sputtering of graphite target was used as the carbon source. The incorporation of Ti and N elements was accomplished by using radio frequency (RF) magnetron sputtering of three different targets: Ti, TiC and TiN, to optimize the mechanical and tribological performance. This work is aimed to clarify the modification mechanism of Ti and N incorporation and obtain the optimum scheme. The influence of RF power on surface topography, chemical composition, mechanical properties and tribological properties was investigated by SEM, XPS, Raman spectra, nanoindentor and tribometer. The consequences revealed that the characteristics of films depend on RF target types and power. For the Ti-C and TiC-C series, when RF power is 100 W and below, with low content of Ti (6 at.%~13 at.%) and N (around 10 at.%), the incorporation of Ti and N optimizes the surface topology, improves the mechanical properties and maintains excellent adhesion to NBR substrate. The tribological and wear behaviors of (Ti:N)-GLC films are better than GLC films under mixed and boundary lubrication. When RF power grows to 200 W, the dopants result in the deterioration of surface and mechanical properties, followed by worse lubrication and wear behaviors. For TiN-C series, the incorporation of TiN takes no advantage over GLC films, even worse in the case of high RF power. Overall, the incorporation of Ti or TiC by magnetron sputtering in Ar/N2 atmosphere is an effective modification method for GLC films on NBR to improve mechanical and tribological behaviors.

**Keywords:** graphite-like carbon (GLC); titanium and nitrogen doping; magnetron sputtering; lubrication; wear

## 1. Introduction

In recent decades, water lubrication bearing has attracted much attention in the marine equipment field owing to its environment-friendly properties. However, the lubrication is poor under mixed and boundary lubrication, as the result of the low viscosity of water, which contributes to intense friction and severe wear. To improve the mechanical and tribological properties of polymers, carbon materials such as carbon nanotubes, graphene, amorphous carbon, etc., have been incorporated in the form of fillers or composite coatings [1–5]. To a great degree, amorphous carbon films deposited on counterparts have been reported as an efficient approach to improving mechanical and tribological behaviors [6–9]. Amorphous carbon is a carbon phase that shows short-range order and long-rage disorder, and contains carbon atoms either mainly in sp2 or sp3 hybridization, corresponding

to graphite-like carbon (GLC) or diamond-like carbon (DLC), respectively [10–12]. DLC (diamond-like carbon) films coated on rubber have been prepared successfully and performed better than uncoated rubber under dry friction [6,8,13–15]. However, DLC films with hydrogen content are very hard with high internal stress and they are quite sensitive to humidity, which gives rise to delamination and wear failure in water [16]. Unlike DLC, GLC (graphite-like carbon) films are able to remain well lubricated and demonstrate wear resistance in both dry and wet conditions. This adaptive feature is very beneficial to water lubrication systems, especially in mixed and boundary lubrication conditions, where a lack of water and direct dry contact occurs [10,17].

Previous studies have focused on the preparation of hard GLC films on silicon wafers or metallic substrates, tribology mechanism against various counterparts and improvement of service behavior in water [10,17–21]. It is challenging work to deposit GLC films on rubber by magnetron sputtering, as thevariable thermal expansion and great deformation capability of rubber force deposited films to be flexible, adherent and load bearing. Furthermore, it is worth mentioning that titanium (Ti) is proven to be an effective doping element in amorphous carbon due to the newly formed Ti-TiC nanocrystallites dispersed in the amorphous carbon matrix [19,20]. Amorphous carbon films doped with Ti can possess high hardness, good toughness, low internal stress, improved tribological performance and wear resistance in aqueous conditions. Moreover, the incorporation of nitrogen (N) in amorphous carbon will form $CN_x$, contributing to several modifications such as low residual stress, strong adhesion strength and good resistance to corrosion, and promoting the graphitization of the film to lower friction [22–25]. Considering their modification, Ti and N co-doped GLC films deserve abundant investigations in water lubrication and are expected to improve tribology under extreme conditions.

In this work, titanium and nitrogen co-doped GLC ((Ti:N)-GLC) films integrated on nitrile butadiene rubber (NBR) were prepared using Direct current (DC) magnetron sputtering of graphite along with RF magnetron sputtering of Ti, TiC or TiN in $Ar/N_2$ atmosphere. An in-depth study of the influence of RF power on surface topology, chemical composition, mechanical properties and tribological properties was conducted. In general, this work aims to investigate the mechanism of doping elements on mechanical and tribological behaviors of GLC films and find the optimum scheme.

## 2. Experimental Details

### 2.1. Deposition Method

Firstly, 3.5 mm thick nitrile butadiene rubber (NBR) sheets with Shore A hardness of $75 \pm 5$ and surface roughness Ra $\leq 0.4$ (400 nm) were used as substrates in this work. (Ti:N)-GLC films were integrated on NBR using multi-target sputtering of Ti (99.99% purity), TiC (99.9% purity) or TiN (99.9% purity) and graphite (99.99% purity) targets (3″ diameter) in $Ar/N_2$ plasma atmosphere. The magnetrons were connected to direct current (graphite) and radio frequency (Ti, TiC or TiN) targets, respectively. Prior to deposition, a standard cleaning procedure described elsewhere was performed to remove oil and wax fillers in rubber [26]. The chamber was evacuated up to a pressure of $6 \times 10^{-3}$ Pa, followed by an Ar plasma etching pretreatment of rubber at a bias voltage of $-500$ V. The etching lasted for 20 min at 4.0 Pa pressure to enhance adhesion strength of films. Throughout the Ar plasma etching pretreatment, the rubber substrates were heated up to 80 °C (measured by a surface thermometer) and then film deposition started. When the deposition began, the vacuum chamber and rubber substrates were cooled down smoothly by a recirculating water system and the deposition temperature was maintained at around 40 °C in order to generate a dense network of cracks which contributed to the flexibility of amorphous carbon films [26]. An Ar (51 sccm) and $N_2$ (3 sccm) mixture gas flow was supplied as working gas. Throughout the deposition, the working pressure remained constant as 1.0 Pa. (Ti:N)-GLC films were deposited on NBR for 90 min with different RF target power ranging from 50 W to 200 W (50 W, 100 W, 200 W), while the DC power connected to graphite target worked constantly at 200 W. For comparison, pure GLC films were prepared under

similar process conditions, using DC magnetron sputtering of graphite at 200 W in Ar (54 sccm) atmosphere. During the whole deposition, to ensure homogeneity, the NBR substrate rotated itself at a constant velocity of 10 rpm and a negative pulse bias voltage of −150 V was exploited to strengthen adhesion further (duty cycle 60%). The total thickness of (Ti:N)-GLC films was about 2.5 μm. For convenience, the prepared films are divided into four groups according to the RF targets and marked with sputtering targets and power, for example, Ti-C (50), TiC-C (100) and TiN-C (200), etc.

### 2.2. Experiment Methods

The surface morphology of (Ti:N)-GLC films was characterized by a scanning electron microscope (JEOL JSM-7800F SEM). Carbon $sp^2/sp^3$ hybridization content and G, D peak position were investigated by Raman spectra, obtained by LabRAM HR Evolution Raman Spectroscope (HORIBA Jobin Yvon S.A.S., Paris, France) at the excitation wavelength of the 532 nm Ar laser line. The chemical composition and bonding types of the films were acquired using an ESCALAB 250Xi type X-ray photoelectron spectroscopy (ThermoFisher Scientific, Waltham, MA, USA), with a radiation source of Al K$\alpha$ radiation.

Considering the viscoelasticity of the NBR substrate, the Nano-scratch test was replaced by the X-cut method to characterize the adhesion strength of (Ti:N)-GLC films on NBR qualitatively, according to the standard steps described elsewhere [27]. The film peeling around the 'X' mark was observed by a VHX-1000C Super-high magnification lens zoom 3D microscope (Keyence, Osaka, Japan). The evaluation was based on ASTM D3359-97 [28].

The hardness and reduced Young's modulus were obtained using a nanoindentor (Hysitron TI-950, Minneapolis, MN, USA) with a conventional Berkovich intender. Then, 4 mN was set as the maximum load to ensure that the maximum indentation penetration depth is within 10–15% of the film thickness, avoiding the influence of viscoelasticity of NBR. For each film, six evenly distributed sites were indented, and the results were averaged.

Tribological tests of (Ti:N)-GLC films were conducted by a tribometer (MFT-5000, Rtec Ltd., San Jose, CA, USA) with a block-on-ring configuration, as shown in Figure 1. The coated NBR (size 15.5 mm × 6.0 mm × 4.0 mm) was bonded to the stator as the block and a dynamic ring (Cr15 steel, roughness Ra 0.2, size $\phi$35 mm × 10 mm) was set as the rotor. The bottom of the ring was immersed in the distilled water and an injector that dripped water at a certain frequency was bundled with the block to ensure the abundance of lubricant. All friction circles were accomplished under the same environmental conditions (25 °C, relative humidity 35%). The load 5 N and 10 N was applied, and the rotational speed was 400 rpm. Each test was performed for 30 min. The worn surface morphology and profile of wear tracks were obtained by a 3D microscope and white light interferometer. The specific wear rate of the worn surface was calculated by the following equation: $K = V/SF$, where $S$ is the friction stroke, $F$ is the applied load and $V$ is the wear volume determined by integrating the cross-sectional profile of the wear track.

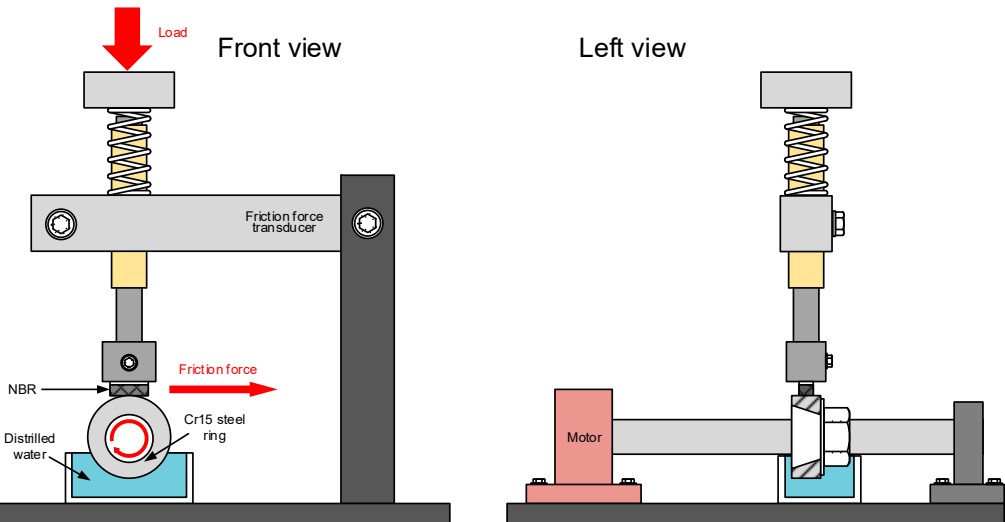

**Figure 1.** Schematic image of block-on-ring tribometer.

## 3. Results and Discussion

### 3.1. Surface Morphology and Topography

The overview of SEM images of surface morphology of (Ti:N)-GLC films on NBR are presented in Figure 2. At low magnification, it is shown that all (Ti:N)-GLC films are divided into micro-scale patches by gaps and ridges, attributed to the mismatch thermal strain of NBR and GLC films. This micro-patched structure is beneficial to internal stress release and improves the flexibility of films deposited on polymers [26]. Figure 3a–c shows the patches of films at 100 W in details: (1) the gaps are closed and perpendicular to the ridges, while few defects (cracks and debris) arise nearby; (2) there are still other patches under the gaps, indicating the staggered growth; (3) the patches are relatively flat and complete. These differences prove that doping of Ti and N improves the toughness of GLC films. The formation of closed gaps and ridges is directly related to the cooling-down process during the deposition. Furthermore, sharp cracks and scattered debris are found near the gap-ridge networks on Ti-C (200), TiC-C (200) and TiN-C (200) films, as is shown in Figure 3d–f, indicating greater internal stress and relatively poor toughness. These defects may arise from the brief and rapid heating and cooling of films caused by particle bombardment during exhaust. Additionally, they tend to peel off under pressure, leading to bad resistance to the erosion and immersion of water. At high magnification, as shown in Figure 4, the surface morphology of (Ti:N)-GLC films exhibit cauliflower-like microstructure, indicating a columnar growth of films, different from the worm-like microstructure of GLC films. As RF target power increases, the microstructure becomes denser, with worse uniformity of clusters.

### 3.2. Bonding Structure and Chemical Composition

Figure 5 shows typical Raman spectra of (Ti:N)-GLC films. The spectra all display a typical feature of amorphous carbon: a significant peak centered around 1570 cm$^{-1}$ with an obvious shoulder peak. To explore the effect of Ti and N incorporation on the carbon hybridization, all the spectra were deconvoluted by the Gaussian function. Generally, the Raman spectra of films consist of two main Gaussian peaks: the strongest one around 1570 cm$^{-1}$ is labeled as the 'G' peak and the shoulder at about 1370 cm$^{-1}$ is labeled as the 'D' peak commonly [29]. It has been widely acknowledged that the G peak represents the bond stretching of sp$^2$ atoms in both aromatic rings and chains, while the D peak originates from the breathing modes of sp$^2$ atoms only in aromatic rings [29]. It must be mentioned that another sub-peak centered at 1450 cm$^{-1}$ attributes to the aromatic substituent in NBR [30]. It is confirmed that the G peak position and intensity ratio of the G peak to the D peak ($I_D/I_G$) can characterize the carbon hybridization ratio (sp$^2$/sp$^3$) qualitatively,

they both increase with the $sp^2/sp^3$ ratio normally. The $I_D/I_G$ and G peak position is illustrated in Figure 5d. For the Ti-C series, the G peak position shifts to a higher wave number with RF power, while the $I_D/I_G$ first decreases from 2.11 to 2.04 and then grows to 3.22, implying more $sp^2$ clusters. For the TiC-C series, the G peak position shifts slightly to lower wave numbers, while the $I_D/I_G$ ratio presents the maximum value as 2.20 at 100 W and is always close to 2.11 of GLC films, which indicates little effect of doped TiC and N on the amorphous carbon bonding. For the TiN-C series, the G peak position is always around 1580 cm$^{-1}$, while the $I_D/I_G$ ratio grows with RF power, from 2.00 to 3.07, implying a more graphite-like structure.

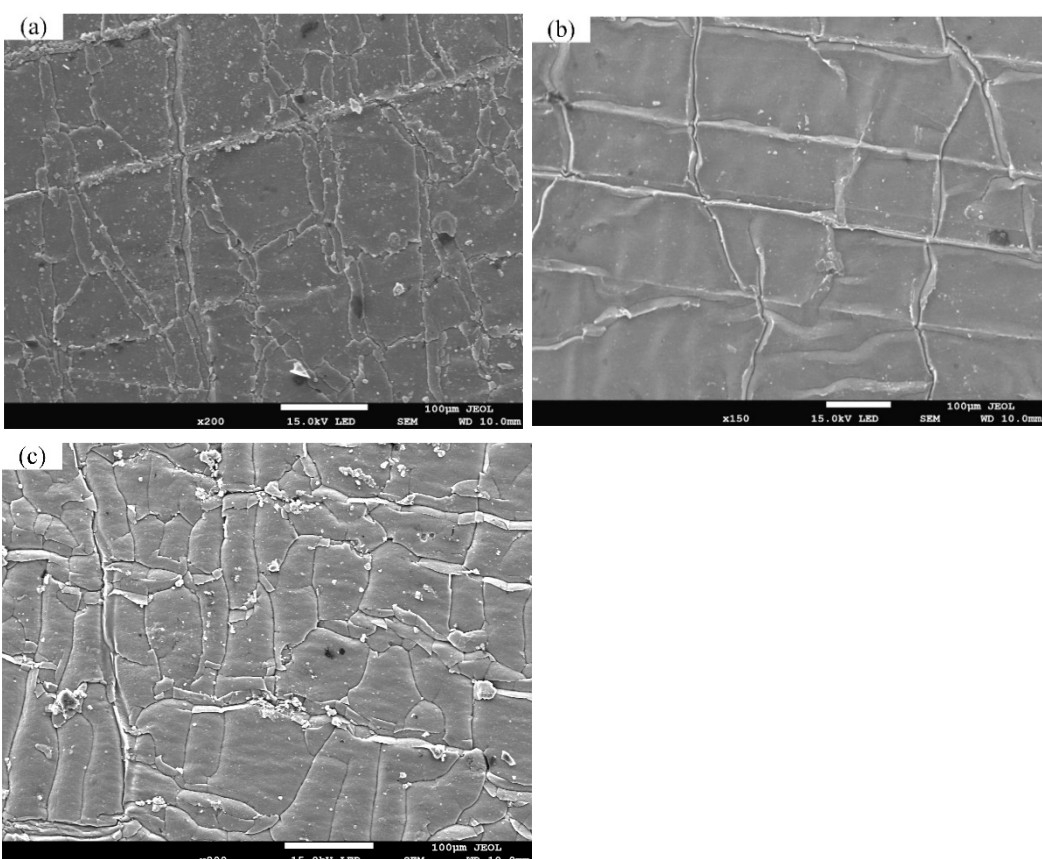

**Figure 2.** SEM images show the overview of patch-like networks of (Ti:N)-GLC films: (**a**) Ti-C (100); (**b**) TiC-C (100); (**c**) TiN-C (100).

The chemical compositions of (Ti:N)-GLC films were taken by XPS. Table 1 shows the content of C, Ti, N and O in (Ti:N)-GLC films. For the Ti-C series, as the Ti target power increases from 50 W to 200 W, the Ti content increases from 5.6 at.% to 17 at.% with a decrease of C content from 69 at.% to 54 at.%. The N content increases from 10.4 at.% to 12 at.%, indicating the formation of TiN or Ti(C,N), which will be discussed by XPS analysis below. For the TiC-C series, with the increase of TiC target power, the C content in films decreases from 69 at.% to 62.4 at.% and N content decreases from 10.7 at.% to 7.9 at.%, while the Ti content increases from 8.5 at.% to 16.2 at.%. This indicates that the dopant TiC may inhibit N incorporation. For the TiN-C series, when the TiN target power increases from 50 W to 200 W, the C content and N content both decrease, from 61 at.% to 52.4 at.% and from 20.4 at.% to 16.3 at.%, respectively, meanwhile the Ti content increases from 8 at.% to 17.7 at.%. The decreasing N content in films indicates less C-N bonding.

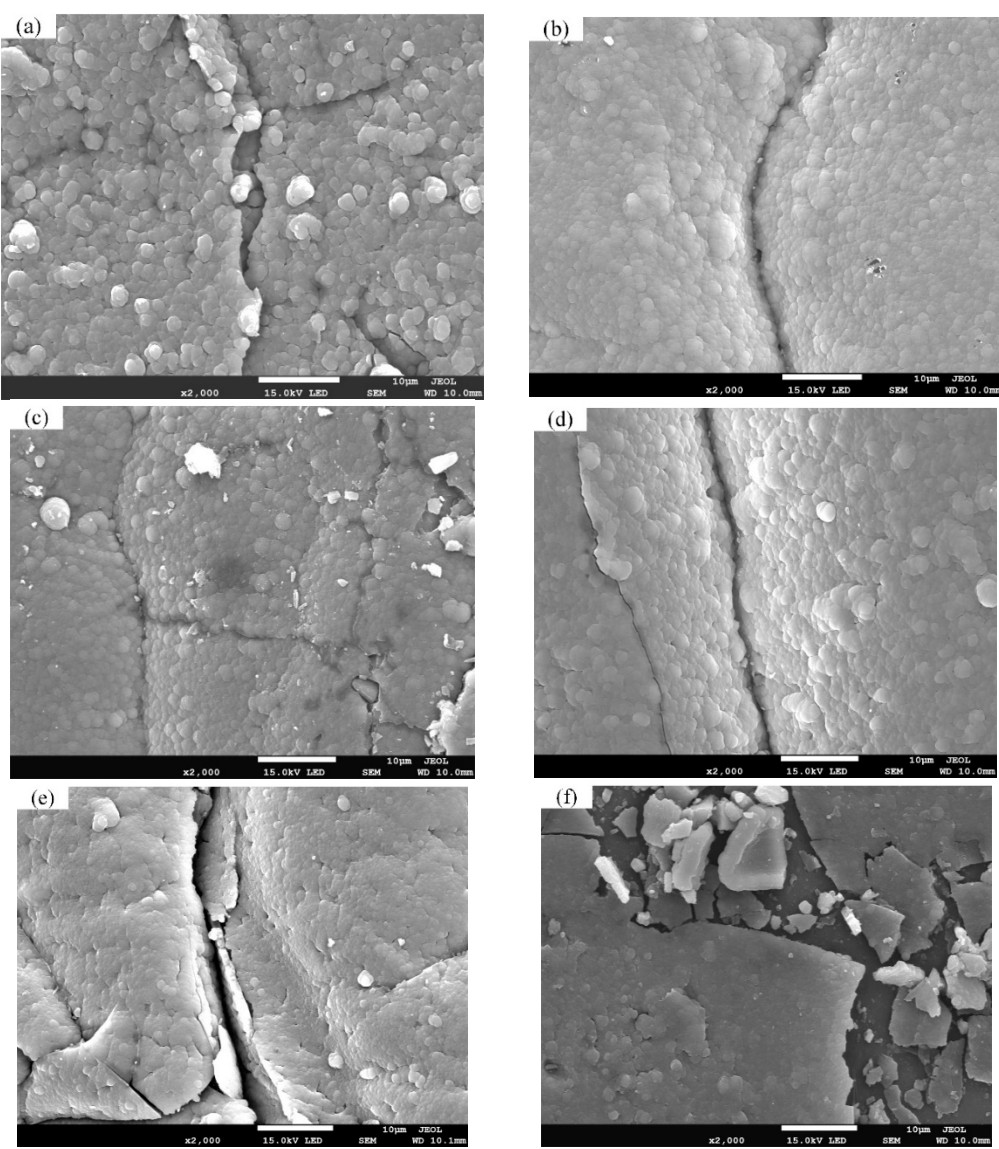

**Figure 3.** SEM images of surface morphology and micro-defects of (Ti:N)-GLC films: (**a**) surface morphology of Ti-C (100); (**b**) surface morphology of TiC-C (100); (**c**) surface morphology of TiN-C (100); (**d**) micro-defects of Ti-C (200) show cracks; (**e**) micro-defects of TiC-C (200) show sharp cracks; (**f**) mico-defects of TiN-C (200) show cracks and debris.

**Table 1.** Chemical composition of (Ti:N)-GLC films.

| RF Target | Power (W) | C (at.%) | Ti (at.%) | N (at.%) | O (at.%) |
|-----------|-----------|----------|-----------|----------|----------|
| **Ti** | 50 | 68.4 | 5.6 | 10.4 | 15.6 |
| | 100 | 59.0 | 12.5 | 11.0 | 17.5 |
| | 200 | 54.3 | 17.0 | 12.0 | 16.7 |
| **TiC** | 50 | 69.0 | 8.5 | 10.7 | 10.8 |
| | 100 | 64.9 | 13.0 | 9.0 | 12.1 |
| | 200 | 62.4 | 16.2 | 7.9 | 13.5 |
| **TiN** | 50 | 61.0 | 8.0 | 20.4 | 10.6 |
| | 100 | 58.0 | 10.0 | 20.2 | 11.8 |
| | 200 | 52.4 | 17.7 | 16.3 | 13.6 |

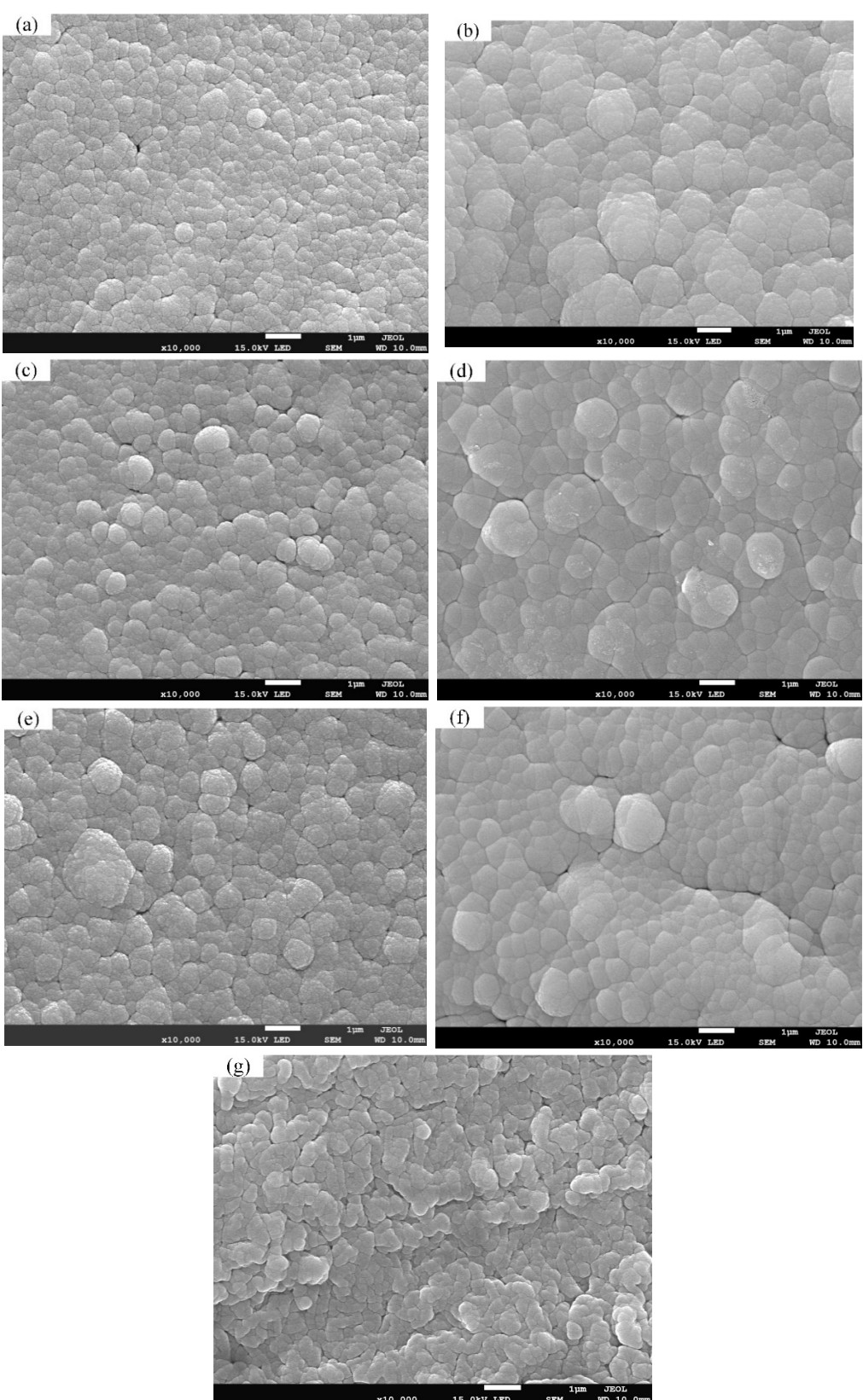

**Figure 4.** SEM images show microstructure of (Ti:N)-GLC films: (**a**) Ti-C (100); (**b**) Ti-C (200); (**c**) TiC-C (100); (**d**) TiC-C (200); (**e**) TiN-C (100); (**f**) TiN-C (200); (**g**) C (200). (Ti:N)-GLC films all exhibit cauliflower-like microstructure, indicating a columnar growth of films, different from the worm-like microstructure of GLC films.

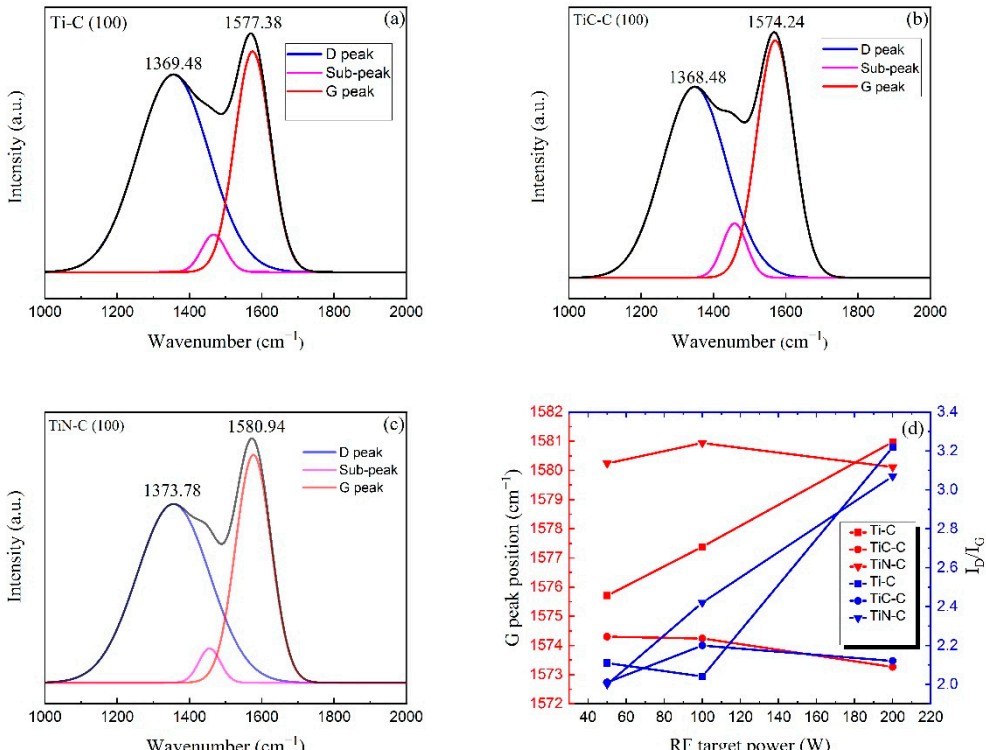

**Figure 5.** Raman analysis of (Ti:N)-GLC films: (**a**) Ti-C (100) spectrum; (**b**) TiC-C (100) spectrum; (**c**) TiN-C (100) spectrum; (**d**) G peak position with $I_D/I_G$ vs. various RF target power.

For detailed chemical bonding of (Ti:N)-GLC films, XPS analysis was taken. To acquire the content of each bonding type, the XPS results were deconvoluted with the Gaussian function. The deconvolution results are illustrated in Figure 6 and Tables 2–4. For the Ti-C series, the strongest peaks at 458.6 eV/464.4 eV can correspond to Ti 2p3/2 and Ti 2p1/2 peak of $TiO_2$, respectively [31], which is proved by the O 1s peak of O-Ti at 529.3 eV observed in the spectra of O 1s. Considering the $Ar/N_2$ atmosphere, the conclusion can be drawn that Ti atoms existed primarily as a solid solution in an amorphous carbon matrix and got oxidized once the films were exposed to air over a few days. Moreover, the peaks at 454.8 eV/460.6 eV in Ti 2p spectra are attributed to TiC, and the last couple of peaks at 456.9 eV/462.9 eV are attributed to Ti(C,N) [32,33]. The new-formed TiC derives from the higher bombardment energy of Ti atoms at higher RF power. Additionally, the formation of Ti(C,N) could be inferred that a certain number of carbon atoms exist in the interstices of TiN formed earlier, since Ti atoms prefer to bonding with N rather than C atoms due to the lower Gibbs free energy needed [33]. N 1s spectra of Ti-C series in Figure 6b are fitted with four sub-peaks: 396.9 eV, 398.0 eV, 399.1 eV and 400.8 eV, representing Ti(C,N), C-N, C=N, and N-O [33]. The C 1s spectra in Figure 6c consist of five Gaussian decomposition peaks: 282.0 eV, 284.0 eV, 284.8 eV, 286.0 eV and 288.0 eV, representing C-Ti, C=C, C-C, C=N and C-O, respectively [31,33–35]. The fitted C-Ti confirms that Ti atoms tend to bond with C atoms at higher RF power, consistent with the peaks of TiC shown in Ti 2p spectra. Furthermore, the carbon $sp^3/sp^2$ ratio is reduced from 0.20 to 0.14, much below that of GLC films at 0.27. Concurrently, the peak area ratio of Ti(C,N) and TiC increases from 0.11 to 0.36. This indicates that more incorporation of Ti atoms at higher power will form more carbides and carbonitrides and break some $sp^3$ bonds to form a more $sp^2$ bonded phase, consistent with the Raman analysis mentioned above. For the TiC-C series, Ti 2p spectra can be deconvolved into three couples of sub-peaks as follow: 454.6 eV/460.6 eV, 455.9 eV/462.0 eV and 458.6 eV/464.6 eV. The peaks at 458.6 eV/464.6 eV are owing to the Ti-O bond and another couple of peaks at 454.6 eV/460.6 eV belong to TiC [34]. The last couple of broad peaks at the binding energy of 455.90 eV/462.0 eV could be inferred as nonstoichiometric Ti*-C [31]. The N 1s spectra of the TiC-C series can be deconvolved

into three peaks centered around 397.9 eV, 399.1 eV and 402.0 eV, regarded as C-N, C=N and N-O, implying the nonbonding effect of TiC on N atoms. The C 1s spectra of TiC-C series are fitted with six peaks: 282.0 eV, 282.8 eV, 284.0 eV, 284.8 eV, 286.0 eV and 288.0 eV, representing C-Ti, C-Ti*, C=C, C-C, C=N and C-O, respectively [34,36]. The C-Ti* bond may be attributed to the interface phase between the amorphous carbon matrix and doped TiC grains. Unlike ones in the middle of clusters, Ti atoms at the clusters' interface are in contact with C atoms both inside and outside clusters, but only neighbor Ti atoms inside clusters [31]. Moreover, the peak intensity ratio of C-Ti* to C-Ti is reduced from 0.68 to 0.42 as RF power grows, indicating the larger grain size of TiC at higher RF power, as is reported by Lewin et al. [34]. For the TiN-C series, the Ti 2p spectra consist of three couples of sub-peaks as follow: 455.5 eV/461.4 eV, 457.0 eV/463.1 eV and 458.6 eV/464.5 eV, recognized as TiN, Ti-N-O and Ti-O, respectively [32,33,35]. The N 1s spectra of the TiN-C series consist of five peaks: 400.7 eV, 399.2 eV, 398.1 ± 0.1 eV, 397.2 ± 0.1 eV and 396.0 eV, corresponding to N-O, C=N, C-N, Ti-N and Ti-N-O. Additionally, Ti-N-O bonding can be confirmed by O 1 s peak around 531.4 eV. It must be mentioned that no bonding of Ti-(C,N) was detected. The C 1s spectra of the TiN-C series can be fitted with four couples of sub-peaks at 284.0 eV, 284.8 eV, 286.2 eV and 288.2 eV, representing C=C, C-C, C=N and C-O, respectively.

**Table 2.** XPS analysis results of Ti 2 p.

| Power (W) | | Total Volume Fraction of Bonds (%) | | | | | |
|---|---|---|---|---|---|---|---|
| Bonds | | Ti-O | TiC | Ti(C,N) | Ti*-C | TiN | Ti-N-O |
| Peak Position/eV | | 458.6/464.5 ± 0.1 | 454.7 ± 0.1/460.6 | 456.9/462.9 | 455.90/462.0 | 455.5/461.4 | 457.0/463.1 |
| Ti | 50 | 88.9 | 3.6 | 7.5 | | | |
| | 100 | 80.3 | 6.2 | 13.5 | | | |
| | 200 | 64.1 | 20.5 | 15.4 | | | |
| TiC | 50 | 5.2 | 74.5 | | 20.3 | | |
| | 100 | 8.7 | 70.1 | | 21.2 | | |
| | 200 | 14.4 | 68.8 | | 16.8 | | |
| TiN | 50 | 8.1 | | | | 64.5 | 27.4 |
| | 100 | 10.9 | | | | 74.3 | 14.8 |
| | 200 | 11.2 | | | | 75.8 | 13.0 |

**Table 3.** XPS analysis results of C 1 s.

| Power (W) | | Total Volume Fraction of Bonds (%) | | | | | |
|---|---|---|---|---|---|---|---|
| Bonds | | C-Ti | C-Ti* | C=C | C-C | C=N | C-O |
| Peak Position/eV | | 282.0 | 282.8 | 284.0 | 284.8 | 286.1 ± 0.1 | 288.1 ± 0.1 |
| Ti | 50 | 2.1 | | 73.9 | 19.9 | 1.2 | 2.9 |
| | 100 | 3.9 | | 77.5 | 15.5 | 1.5 | 1.6 |
| | 200 | 5.3 | | 76.3 | 10.7 | 4.6 | 3.1 |
| TiC | 50 | 4.5 | 3 | 68.8 | 17.5 | 4.9 | 1.3 |
| | 100 | 11.4 | 5.7 | 63.3 | 15.8 | 3.2 | 0.6 |
| | 200 | 14.3 | 6.0 | 59.5 | 14.9 | 1.8 | 3.5 |
| TiN | 50 | | | 73.9 | 18.4 | 5.7 | 2.0 |
| | 100 | | | 73.0 | 19.0 | 5.8 | 2.2 |
| | 200 | | | 82.6 | 14.0 | 1.7 | 1.7 |

**Table 4.** XPS analysis results of N 1 s.

| Power (W) | | Total Volume Fraction of Bonds (%) | | | | | |
|---|---|---|---|---|---|---|---|
| **Bonds** | | **Ti-N-O** | **Ti(C,N)** | **Ti-N** | **C-N** | **C=N** | **N-O** |
| **Peak Position/eV** | | **396.0** | **396.9** | **397.2 ± 0.1** | **398.0 ± 0.1** | **399.1** | **400.7 402.0** |
| **Ti** | 50 | | 6.3 | | 19.6 | 72.5 | 1.6 |
| | 100 | | 6.3 | | 20.3 | 69.9 | 3.5 |
| | 200 | | 8.1 | | 24.0 | 58.5 | 9.4 |
| **TiC** | 50 | | | | 21.2 | 71.0 | 6.8 |
| | 100 | | | | 25.9 | 68.0 | 6.1 |
| | 200 | | | | 42.8 | 51.5 | 5.7 |
| **TiN** | 50 | 2.1 | | 11.3 | 14.4 | 63.6 | 8.6 |
| | 100 | 6.8 | | 27.6 | 11.3 | 45.2 | 9.1 |
| | 200 | 4.3 | | 39.2 | 13.7 | 36.5 | 6.3 |

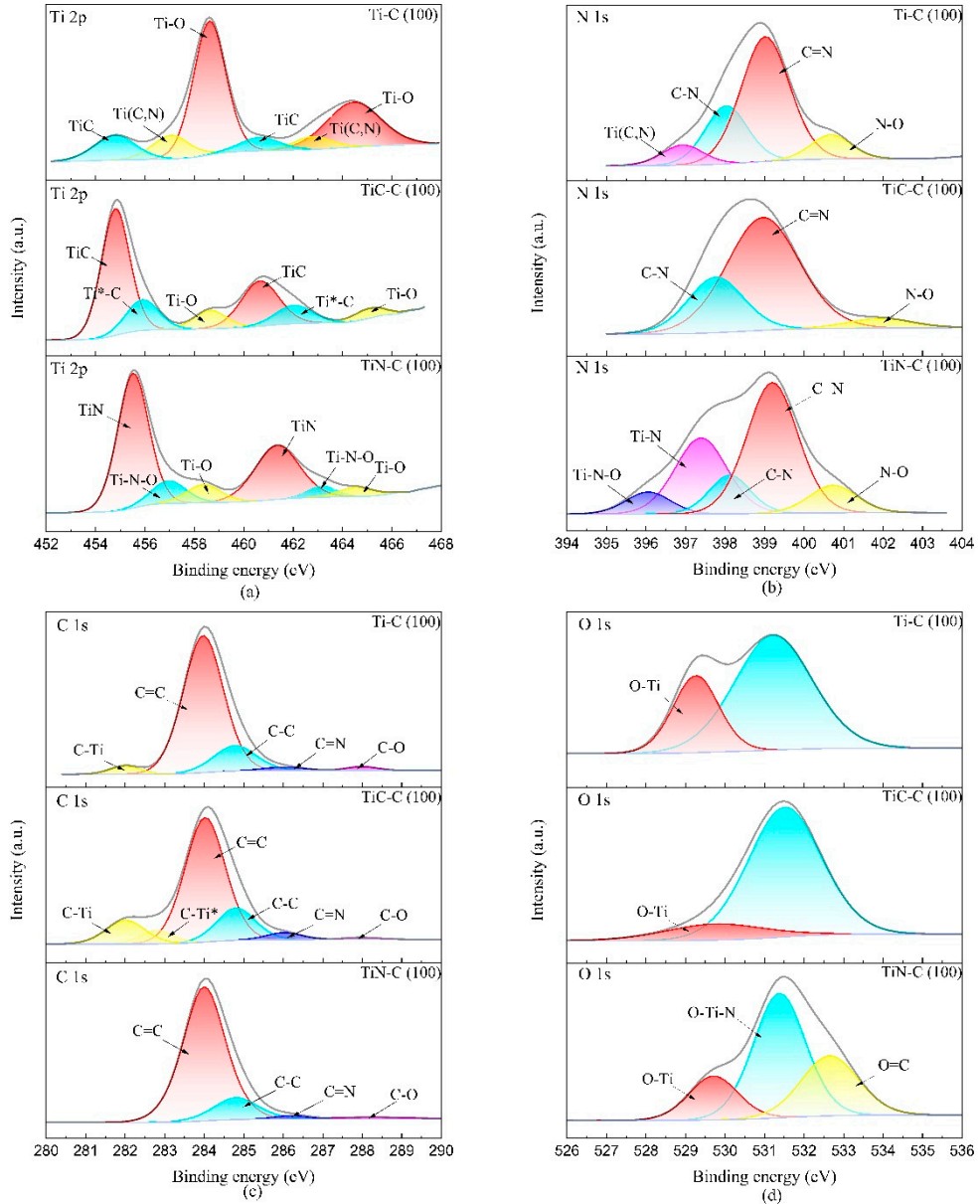

**Figure 6.** XPS analysis of (Ti:N)-GLC films: (**a**) Ti 2p; (**b**) N 1s; (**c**) C 1s; (**d**) O 1s.

### 3.3. Mechanical Properties

To determine the adhesion levels qualitatively, the X-cut method was carried out. Additionally, the evaluation was based on ASTM D3359-97 [28]. The optical microscopy images of the 'X' mark and adhesion levels are shown in Figure 7 and Table 5. It could be observed that, for TiN-C (100) and GLC films, the brittle fracture took place and led to jagged removal along incisions during tape peeling, indicating relatively poor adhesion (3A level). For TiC-C (100) and Ti-C (100) films, trace peelings occurred along incisions, which could be determined as 4A level [6,27]. Moreover, the adhesion levels all dropped to 3A or worse level at 200 W.

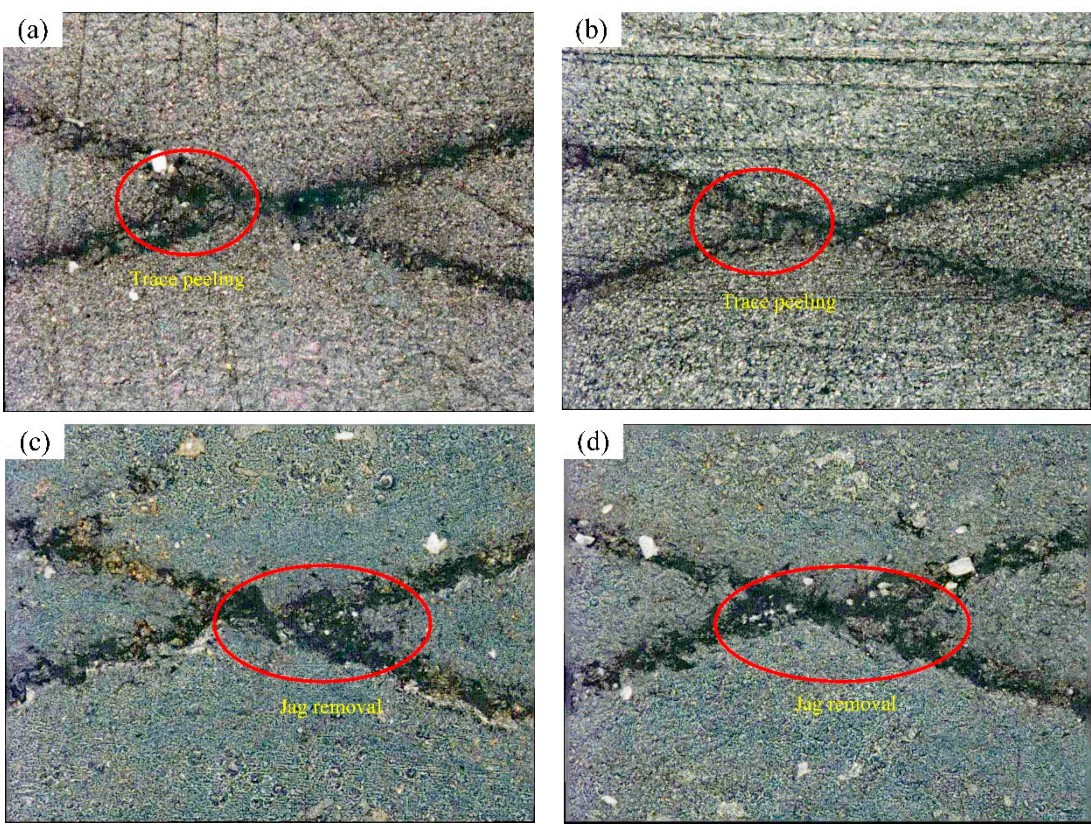

**Figure 7.** Optical microscopy images of the 'X' mark on (Ti:N)-GLC films: (**a**) Ti-C (100); (**b**) TiC-C (100); (**c**) TiN-C (100); (**d**) GLC.

**Table 5.** Film adhesion ratings according to ASTM D3359-97.

| Adhesion Level | Performance |
| --- | --- |
| 5A | No peeling or removal occurs at all (absence of peeling) |
| 4A | Trace peeling or removal occurs along incisions (no peeling occurs at the intersect and little peeling observed at the X-cut) |
| 3A | Jagged removal along incisions occurs up to 1/16 in. (1.6 mm) on either side of the intersect of the X-cut |
| 2A | Jagged removal along incisions occurs up to 1/8 in. (3.2 mm) in either direction from the intersect of the X-cut |
| 1A | Most of the X-cut area peeled off with the adhesive tape |
| 0A | Removal beyond the X-cut area occurs |

Nanoindentation was carried out to investigate the hardness (*H*) and reduced Young's modulus (*E*) of (Ti:N)-GLC films. Nanoindentation curves of (Ti:N)-GLC films are shown in

Figure 8. Additionally, the results were calculated with reference to the standard Oliver and Pharr approach [37], and listed in Figure 9a. For the Ti-C series, the incorporation of Ti and N exerts an impressive influence on the mechanical properties of films. When RF power is below 200 W, the strengthening effect on mechanical properties could be attributed to commonly formed $TiO_2$, TiC and Ti(C,N) phases, which have good mechanical properties and relatively little effect on the connectivity of amorphous carbon matrix and carbon $sp^3$ content. However, as RF power grows, the balance is broken, and more and larger dopant groups break the connectivity of the amorphous carbon structure and promote carbon phase transition from $sp^3$ to $sp^2$, eventually reducing hardness and leading to a reduced Young's modulus. For TiC-C series, the (Ti:N)-GLC films all perform better mechanical properties than GLC films, attributed to the good compatibility of Ti carbides with carbon and the stable $sp^2/sp^3$ ratio mentioned above. The highest hardness ($H$ = 12.9 GPa) arises at 100 W and the highest value of the reduced Young's modulus ($E$ = 144.4 GPa) is at 200 W. For the TiN-C series, when the RF power is 100 W and below, the mechanical properties of (Ti:N)-GLC films are both better than those of GLC films, and the highest values ($H$ = 10.2 GPa and $E$ = 139.9 GPa) are obtained at 100 W. However, as RF power grows further, the mechanical properties of films degenerate rapidly, even worse than those of GLC films. In general, the hardness ($H$) and reduced Young's modulus ($E$) of (Ti:N)-GLC films in this work ($H$ = 7.9 GPa~12.9 GPa, $E$ = 120.9 GPa~144.4 GPa, respectively) are comparable to those ($H$ = 8.9 GPa~11.6 GPa, $E$ = 122.0 GPa~153.6 GPa, respectively) of Ti-containing GLC films obtained by Wang et al. [20].

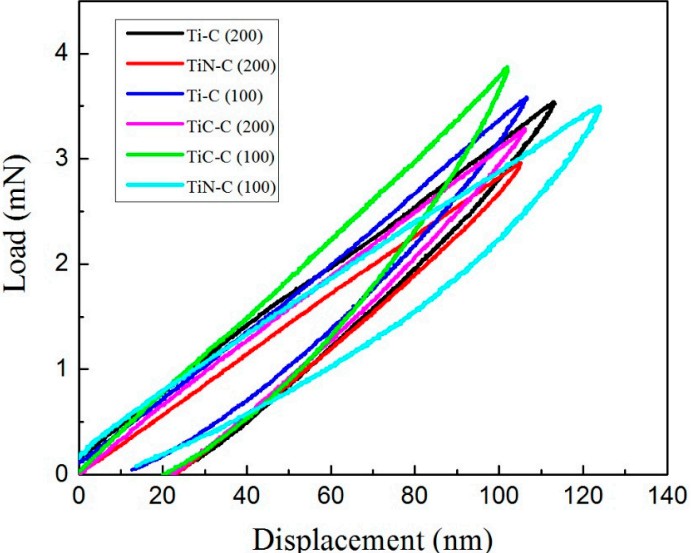

**Figure 8.** Nanoindentation load vs. indentation displacement curves of (Ti:N)-GLC films.

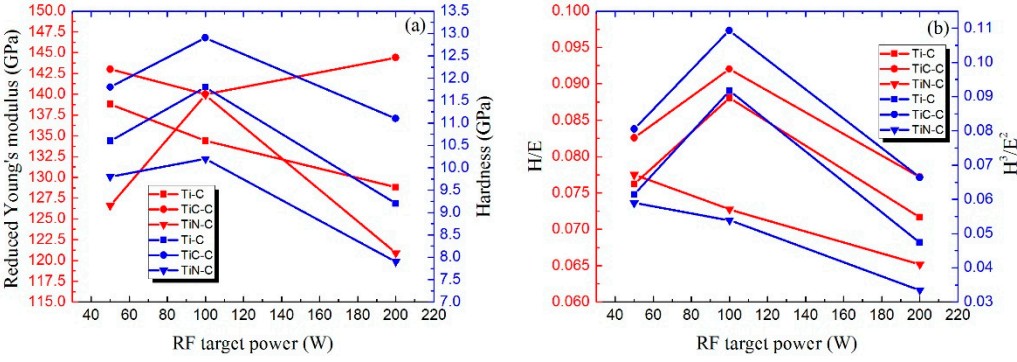

**Figure 9.** Mechanical properties of (Ti:N)-GLC films: (**a**) Hardness (*H*) and Reduced Young's modulus (*E*); (**b**) *H/E* ratio and $H^3/E^2$ ratio.

It is widely confirmed that the $H/E$ and $H^3/E^2$ ratios are closely relevant to the elastic strain to failure and resistance to plastic deformation of films, respectively [36,38], considered indicators of the wear resistance performance and toughness of films, they are both calculated and listed in Figure 9b. Considering these two ratios, TiC-C films and Ti-C (100) films display excellent mechanical properties. Concurrently, Ti-C (200) and TiN-C (200) films perform worse, consistent with the poor toughness shown above in Figure 3d–f.

### 3.4. Friction Behavior

Static contact angle measurement was conducted to confirm the wettability of (Ti:N)-GLC films. As is shown in Figure 10, the modified NBR surface performs a greater contact angle and becomes hydrophobic. For Ti-C and TiC-C series, the contact angle increases with RF power, indicating that the hydrophilic surface becomes hydrophobic. At the same time, TiN-C films appear to be hydrophobic all the time and the contact angle increases with RF power.

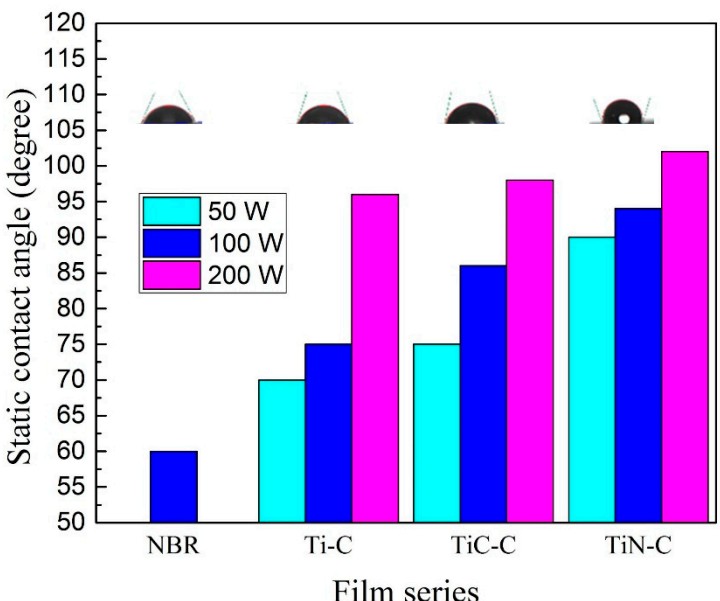

**Figure 10.** Static contact angle of (Ti:N)-GLC films.

Figure 11a illustrates the friction coefficient (CoF) curves of (Ti:N)-GLC films (100 W) under mixed lubrication, and the mean steady-state friction coefficients at various RF power are described in Figure 11b. For TiC-C (100) film and Ti-C (100) film, the friction coefficient is maintained in a low stable range of 0.049~0.070, much below that of NBR (0.154) and GLC film (0.095). Concurrently, for TiN-C (100) film, the friction coefficient keeps increasing and exceeds GLC film after about 8200 circles, indicating the deterioration of lubrication. As can be seen in Figure 11b, when RF power grows, the mean steady-state friction coefficients of the TiN-C series and Ti-C series gradually increase (0.080 to 0.104 and 0.069 to 0.087, respectively), while the TiC-C films own the lowest value 0.061 at 100 W and highest value as 0.075 at 200 W.

Figure 12a shows the friction coefficient (CoF) curves of (Ti:N)-GLC films (100 W) under boundary lubrication. Figure 12b exhibits the mean steady-state friction coefficients at various RF power. It could be observed that the lubrication of Ti-C (100) film and TiC-C (100) film is stable, and the friction coefficient is below 0.150, while TiN-C (100) film has an obvious running-in phase before 1000 cycles, and the steady-state friction coefficient is close to GLC film near 0.250. Investigating the influence of RF power, the mean steady-state friction coefficients of (Ti:N)-GLC films all increase with RF power.

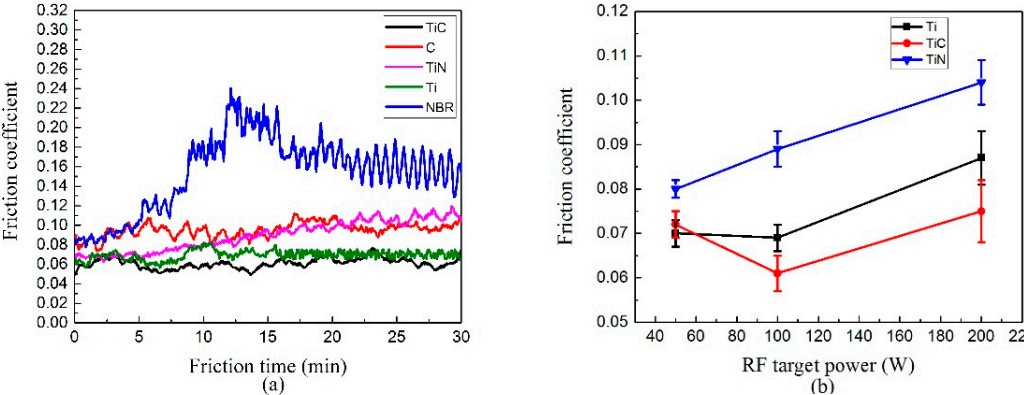

**Figure 11.** Friction coefficient (CoF) of (Ti:N)-GLC films under mixed lubrication: (**a**) friction coefficient curves of films at 100 W; (**b**) the mean steady-state friction coefficient vs. RF power.

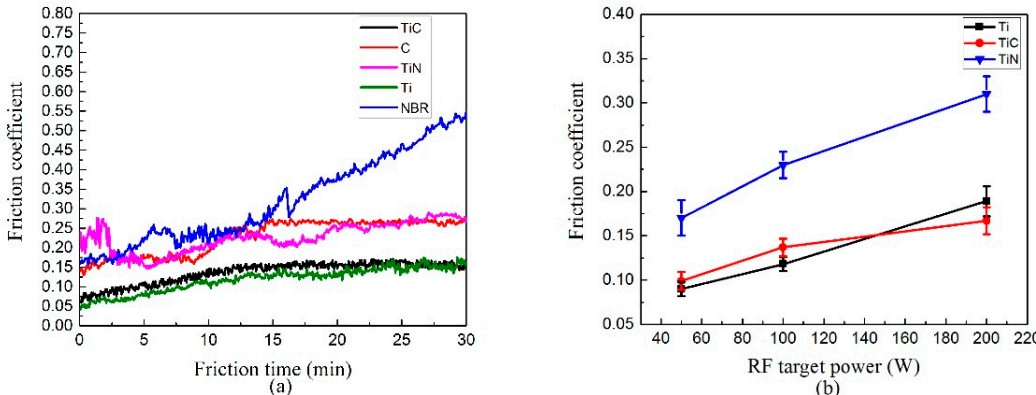

**Figure 12.** Friction coefficient (CoF) of (Ti:N)-GLC films under boundary lubrication: (**a**) friction coefficient curves of films at 100 W; (**b**) the mean steady-state friction coefficient vs. RF power.

To investigate the lubrication mechanism of modified films, Stribeck curves of water lubrication were obtained for (Ti:N)-GLC films (100 W) and GLC films. The Stribeck curves shown in Figure 13 illustrate the correlation between the Sommerfeld number $\eta v / P$ and the mean steady-state friction coefficient, where $\eta$ is viscosity of distilled water, $v$ is line speed of rotator and $P$ is load per unit length. For (Ti:N)-GLC films, the lubrication is determined by two behaviors between counterparts: hydrodynamic lubrication caused by water film and self-lubrication of GLC at dry contact, and the competition between them determines the lubrication state. When the load is 20 N ($\eta v / P = 0.5$), the water film between counterparts cannot bear the load and breaks down, leading to the direct contact between counterparts, where boundary lubrication occurs. The lubrication at direct dry contact depends on the low shear strength of the graphite-like structure and newly formed carbonaceous transfer film, where direct contact between counterparts is prevented and the load-bearing capacity is improved. Furthermore, the hydrophilic TiC-C (100) and Ti-C (100) surfaces are conducive to the formation and survival of very thin fragmented water film on the interface, reducing direct dry contact further to lower the friction coefficient. As the load drops to 5 N ($\eta v / P = 2.0$), the lower load allows the water film to exist on a larger area of interface and the friction coefficient reduces sharply. Nevertheless, the extremely thin water film cannot cover the entire interface, and asperities on counterparts are still in direct contact, leading to mixed lubrication.

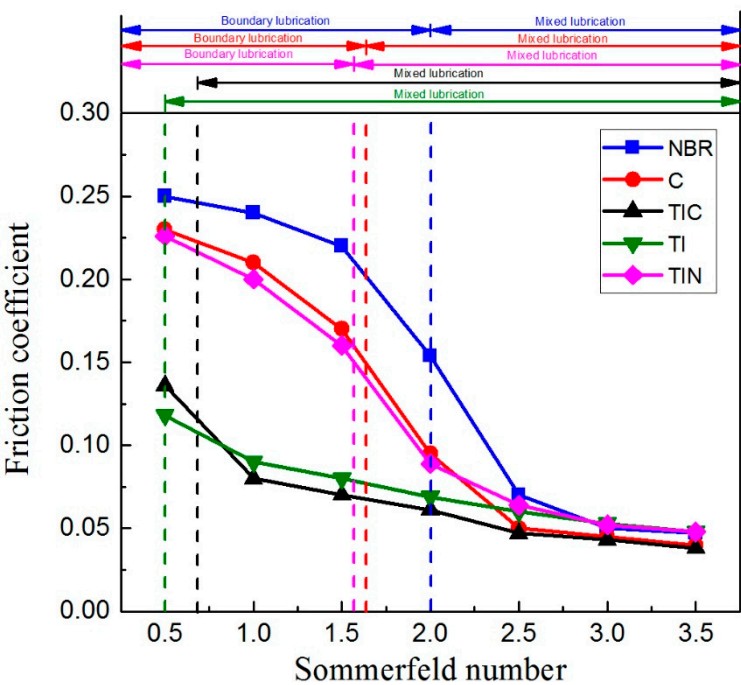

**Figure 13.** Stribeck curves of (Ti:N)-GLC films at 100 W.

Moreover, the influence of RF power on the lubrication mechanism could be explained as the following three aspects: (1) When RF power grows, the size of the dopant groups such as $TiO_2$, $Ti(C,N)$, $TiC$, $TiC_xO_y$ and $TiN$, etc., on the film surface increases, resulting in poor consistency and increased roughness. The bumpy surface produces a large amount of wear debris mainly composed of the dopant groups mentioned above on the lubricating interface during solid-to-solid contact. The wear debris exerts an abrasive effect on the interface, which accelerates the deterioration of the film. (2) As RF power grows, the film becomes hydrophobic, which is hard for the molecular-scale lubricating water film to survive on the surface. (3) As mentioned above, the internal residual stress increases with RF power, and the stress release leads to a significant increase of microscopic defects shown in Figure 3d–f and poor adhesion to the NBR substrate described in Figure 7, weakening the load-bearing capacity of films. Eventually, films deposited at higher RF power are more likely to peel off, resulting in local failure of lubrication. However, for the TiC-C series, thanks to the stability of TiC, no bonding effect with N, and good compatibility with carbon matrix, the film maintains better mechanical properties, so that the load-bearing capacity remains.

*3.5. Wear Behavior*

The calculated specific wear rate of (Ti:N)-GLC films is shown in Figure 14. For the Ti-C series, Ti-C (100) film obtains the lowest wear rate value: $7.52 \times 10^{-8}$ mm$^3$/Nm under mixed lubrication and $18.2 \times 10^{-8}$ mm$^3$/Nm under boundary lubrication. When RF power is 200 W, the wear rate reaches the maximum value at $11.2 \times 10^{-8}$ mm$^3$/Nm and $30.5 \times 10^{-8}$ mm$^3$/Nm, respectively. For the TiC-C series, films reveal a low and stable wear rate: $6.4 \times 10^{-8}$ mm$^3$/Nm~$10.9 \times 10^{-8}$ mm$^3$/Nm under mixed lubrication and $13.6 \times 10^{-8}$ mm$^3$/Nm~$23.7 \times 10^{-8}$ mm$^3$/Nm under boundary lubrication. For the TiN-C series, the wear rate always increases with RF power. Moreover, the wear rate rises sharply to $58.2 \times 10^{-8}$ mm$^3$/Nm with RF power under boundary lubrication, even worse than in GLC films. Overall, the wear rate values of Ti-C and TiC-C films are close, and TiC-C (100) film performs the best. In contrast, the anti-wear performance of TiN-C films is the worst, especially under boundary lubrication.

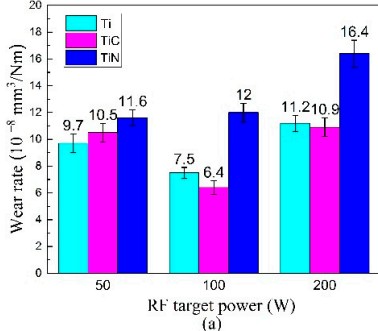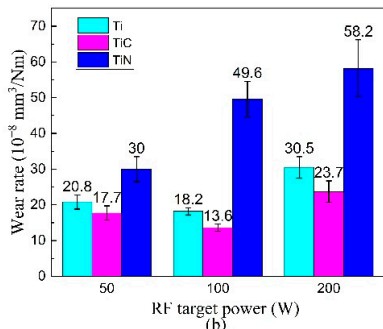

**Figure 14.** Wear rate of (Ti:N)-GLC films: (**a**) Mixed lubrication; (**b**) Boundary lubrication.

To investigate the wear mechanism further, a 3D microscope and white light interferometer were exploited to characterize the morphology of the worn surfaces and the profile of wear tracks. The worn surfaces and wear tracks of Ti-C (100) film under two lubrications are demonstrated in Figure 15a,b, few wear debris could be observed on wear tracks and the wear depth is 0.7 μm and 1.0 μm, respectively. Obviously, the wear tracks are much smoother than the unworn surface, indicating that asperities are polished rather than stripped. In addition, much flaking is observed on the non-contact area under boundary lubrication, indicating fatigue delamination and peeling-off caused by water erosion. As shown in Figure 15c, there are fragmented films on the worn surface, indicating fatigue delamination of TiC-C (100) film under mixed lubrication, and the wear depth is around 0.6 μm. The worn surface under boundary lubrication in Figure 15d shows debris accumulation on both sides of the wear track and micro-scratches in the center, indicating cooperation of adhesive wear and abrasive wear. The inset shows that the wear depth is 0.8 μm. For TiN-C (100) film under mixed lubrication, Figure 15e shows much wear debris on the wear track along with peeling-off on both borders, indicating the combination of fatigue delamination and adhesive wear, and the wear depth is 1.2 μm. Under boundary lubrication, the micro-grooves on the wear track with debris accumulation on both borders in Figure 15f could be ascribed to severe abrasive wear and adhesive wear. The wear depth is 2.9 μm, more than film thickness, indicating the worn-out and lubrication failure of TiN-C (100) film.

Generally speaking, the wear behaviors of (Ti:N)-GLC films are mainly correlated with mechanical properties and interaction with water. Under mixed lubrication, the wear mechanism of the films is mainly fatigue delamination and peeling-off caused by water erosion. As discussed above, when RF power grows, the mechanical properties of the films begin to deteriorate after the inflection point, giving rise to more microscopic defects and poor adhesion to the NBR substrate, which would be easier for water molecules to infiltrate into films. The wear behaviors under boundary lubrication are mainly determined by the cooperation of adhesive wear and abrasive wear. As RF power grows, the dopant groups in films increase and grow, leading to worse mechanical properties and rougher surfaces. Concurrently, the dopant groups in wear debris increase significantly, which aggravates adhesive wear and abrasive wear.

(**a**)

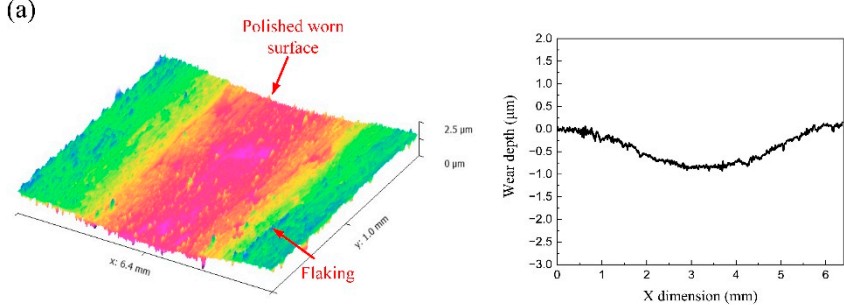

**Figure 15.** *Cont.*

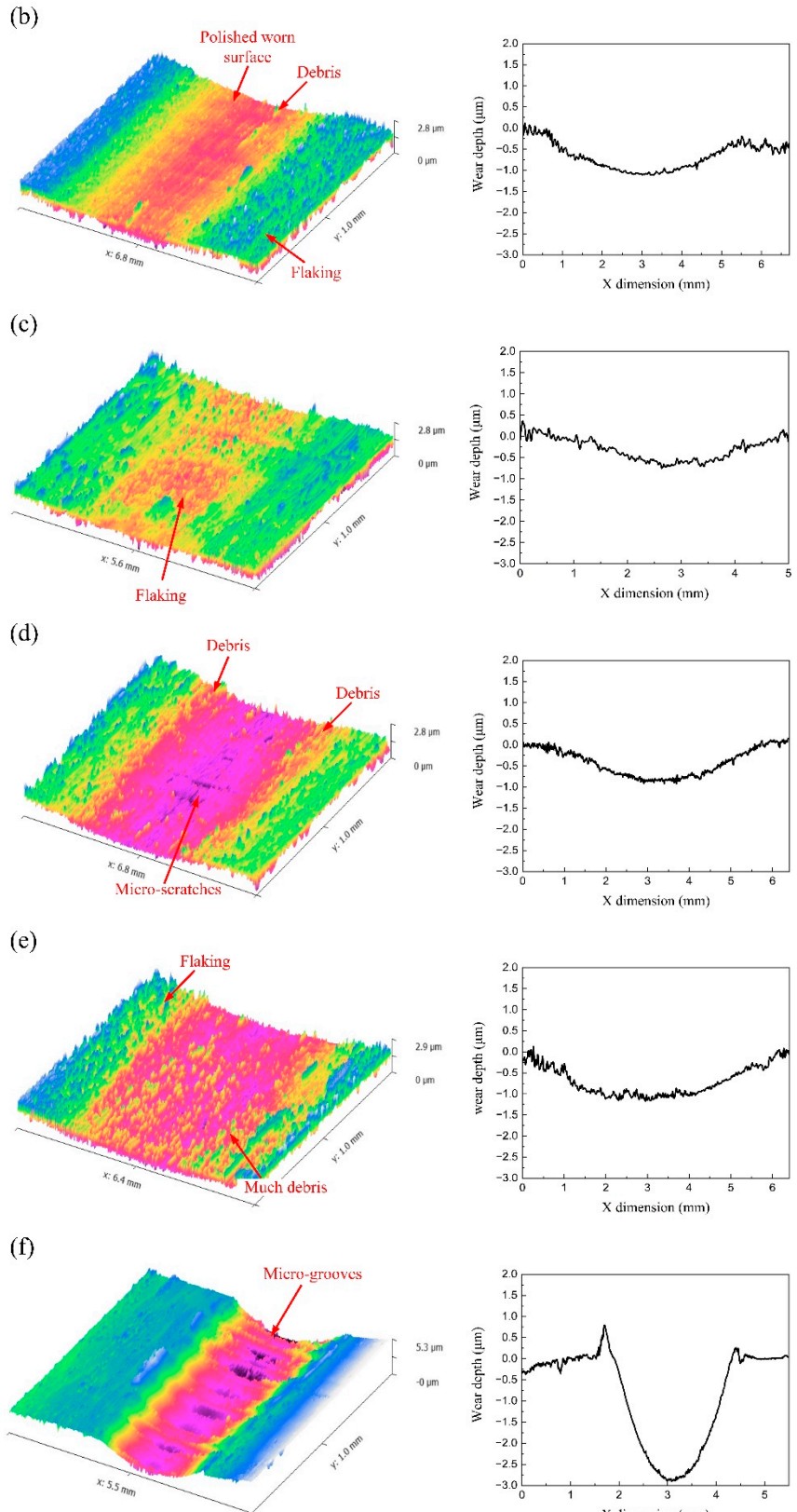

**Figure 15.** Worn surface morphology and profile of wear tracks: (**a**) Ti-C (100) mixed; (**b**) TiC-C (100) mixed; (**c**) TiN-C (100) mixed; (**d**) Ti-C (100) boundary; (**e**) TiC-C (100) boundary; (**f**) TiN-C (100) boundary.

## 4. Conclusions

(Ti:N)-GLC films were successfully deposited on NBR using multi-target magnetron sputtering. RF magnetron sputtering of three different targets: Ti, TiC and TiN in $Ar/N_2$ atmosphere was conducted to incorporate Ti and N elements into GLC films, to optimize the mechanical and tribological performance. For further detailed research and comparison, the influence of RF power on surface topography, chemical composition, mechanical properties and tribological properties was investigated. The following conclusions could be drawn from this work:

(a)  For Ti-C and TiC-C series, at low RF power, the incorporation of Ti and N has little effect on the connectivity of the amorphous carbon matrix. Additionally, newly formed dopant groups like $TiO_2$, TiC and Ti(C,N), etc., have a strengthening effect on the structure of GLC films, thereby optimizing the surface topology, improving the mechanical properties ($H$ = 9.2 Gpa~12.9 GPa, $E$ = 128.8 Gpa~143.0 GPa) and maintaining good adhesion to NBR substrate. In return, better surface and mechanical properties can impressively optimize the lubrication and reduce the wear under mixed (CoF = 0.061~0.072 and $K$ = $6.4 \times 10^{-8}$ $mm^3/Nm$~$10.5 \times 10^{-8}$ $mm^3/Nm$) and boundary (CoF = 0.090~0.137 and $K$ = $13.6 \times 10^{-8}$ $mm^3/Nm$~$20.8 \times 10^{-8}$ $mm^3/Nm$) lubrication.

(b)  For Ti-C and TiC-C series, when RF power grows to 200 W, the dopant groups grow, along with the emergence of more dopant oxides. The connectivity and stability of amorphous carbon structures are destroyed, resulting in the deterioration of mechanical and surface properties. The worse surface and mechanical properties lead to fatigue delamination of films in water and a combination of adhesive and abrasive wear under boundary lubrication. Consequently, the friction coefficient and wear rate both grow under mixed (CoF = 0.075~0.087 and $K$ = $10.9 \times 10^{-8}$ $mm^3/Nm$~$11.2 \times 10^{-8}$ $mm^3/Nm$) and boundary (CoF = 0.152~0.180 and $K$ = $23.7 \times 10^{-8}$ $mm^3/Nm$~$30.5 \times 10^{-8}$ $mm^3/Nm$) lubrication.

(c)  Unlike the Ti-C and TiC-C films, the incorporated TiN hardly bonds to amorphous carbon atoms and survives as a solid solution in the carbon matrix. The dopant TiN and oxide destroy the connectivity and stability of the amorphous carbon structure and grow larger with RF power, which increases the internal stress of the film, resulting in poor adhesion and mechanical properties. More surface defects and degraded mechanical properties make (Ti:N)-GLC films suffer from severe fatigue delamination under water lubrication, meanwhile more wear debris aggravates adhesive and abrasive wear, leading to film failure eventually. Therefore, (Ti:N)-GLC films of the TiN-C series take no advantage over GLC films under water lubrication (CoF = 0.080~0.230 and $K$ = $11.6 \times 10^{-8}$ $mm^3/Nm$~$49.6 \times 10^{-8}$ $mm^3/Nm$), and even worse in the case of higher RF power (CoF > 0.250 and $K$ > $52.0 \times 10^{-8}$ $mm^3/Nm$).

Based on mentioned above, the incorporation of Ti or TiC by magnetron sputtering in an $Ar/N_2$ atmosphere can be an optimum modification method for GLC films on NBR, in terms of mechanical and tribological behaviors especially under mixed and boundary lubrication. Additionally, it must be emphasized that the content of Ti and N should be appropriately low (around 6 at.%~10 at.%).

**Author Contributions:** Z.Z.: Methodology, visualization, writing—original draft, formal analysis. Y.H.: writing—review and editing, funding acquisition. J.Q.: investigation. All authors have read and agreed to the published version of the manuscript.

**Funding:** This research received no external funding.

**Institutional Review Board Statement:** Not applicable.

**Informed Consent Statement:** Not applicable.

**Data Availability Statement:** Data sharing is not applicable to this article.

**Acknowledgments:** The authors gratefully acknowledge the financial support from the National Natural Science Foundation of China (Grant No. 51975064).

**Conflicts of Interest:** The authors declare no conflict of interest.

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
