# Peer review of "Improving Mechanical and Tribological Behaviors of GLC Films on NBR under Water Lubrication by Doping Ti and N"

_coatings, doi:10.3390/coatings12070937_

Round 1
Reviewer 1 Report
Overall, this is an interesting contribution to tribological carbon coatings on rubber against the application background of sealing rings under water lubrication. The approach of the paper to optimize already known tribological GLC coatings by doping is a good approach. The execution of the experiments, the analyses and evaluation methods are reasonable. The paper is basically suitable for publication in this Special issue, but it needs major revision with respect to the following aspects:
General remarks:
The term GLC is common and forms the counterpart to the designation of DLC coatings, which are for predominantly diamond (sp3-) bonded. However, it has become common practice for most researchers to refer to their (amorphous) carbon films as DLC type, regardless of whether the sp3 content is above 50%. In this respect, the authors of this paper are, in a sense, on an island by making almost no reference at all to other work dealing with DLC films that are very similar to the GLC films discussed here. Please note this fact and improve the manuscript as follows:
1) Classify your coatings into the nomenclature of DLC coatings according to ISO 20523:2017 'Carbon based films'. After you have clarified the terminology, you can keep the term (Ti:N)-GLC, etc. in the further.
2) By confining to GLC you make no reference at all to literature work with DLC, although very similar coatings on rubber have been studied here. My recommendation is to look at the following papers and make a reference to your work: doi:10.1016/j.triboint.2004.07.007, doi:10.1016/j.surfcoat.2014.08.016, doi:10.1016/j.surfcoat.2009.01.027, doi:10.1016/j.surfcoat.2011.04.011
Specific remarks:
3) Abstract, lines 18-23: too many details and numbers; I recommend that only general statements be made here and that no concrete numerical values be given
4) Abstract, lines 24-26: This sentence is contradictory to the previous sentence. Please revise.
5) line 44: Better delete the word "highly”
6) lines 46/47: This is not true – see comment 2
7) line 98: replace “presented” by “characterized”
8) lines 151/152, Figure 2: Why do images a, b, and c contain inserted images in the upper right corner?
9) lines 151/152, Figure 2: Why are images d, e, and f shown at a different magnification (than a, b, and c)? Please replace them in the same magnification so that one can directly compare the crack characteristics.
10) lines 151/152, Figure 2: Nowhere the arrows in the images d, e, and f are addressed. Then it is better not to insert them.
11) lines 162-232: The chapter is called Bonding structure and chemical composition - but the chemical composition is not shown and discussed at all!
12) lines 187-232: This paragraph is very confusing and contains a lot of information that a reader cannot process in this way. My recommendation is to structure the approach more strongly and also to work with a table. In this table, the results of the chemical composition (see remark 11) should also be inserted, so that one can see the connection between composition and bond structure.
13) lines 237-243: On base of what parameter the evaluation of adhesion level was done?
14) line 244, Figure 6: I cannot see any delamination. What information can be obtained from the images with respect to my remark 13?
15) line 280: Replace Tribological by Friction (because Tribological includes Friction and Wear) and write Behavior (instead of Behaviors).
16) lines 339/340: Please reformulate this sentence so that it is easier to understand.
17) line 348: Behavior (instead of Behaviors)
18) lines 407-410: The chemical composition results appear here for the first time (see remark 11); The sentence is difficult to understand and contains too much information. Please revise the wording.
19) general comment: Please revise the labels and legends in all figures so that all numbers and letters can be seen clearly. The resolution of the diagrams is also very poor at the moment. A much better quality must be available for publication.
Reviewer 2 Report
The paper is interesting. However, in its current form it cannot be published. The main shortcoming that prevents a full review process is the quality of the figures. Most of the inscriptions and markings are illegible. I would be very grateful if the quality of the drawings could be improved. In addition, the authors propose to use a covering and the special drawings show cracks. Please include in the paper drawings of the Young's modulus measurement tests performed and indicate where exactly they were performed. Because the Young's modulus measurement will very much depend on the place where this measurement is taken. I am referring to the nanoindentation measurement near cracks. Please also comment on the effect of these problems on Young's modulus and microhardness. It would be useful to include graphs of the nanoindentation tests performed.
Furthermore, the question arises whether the introduction of microcracks under wet friction conditions is desirable? Please show the SEM images after the tryboloigic tests and comment in the context of the introduced cracks what effect they had on the surface failure.
Reviewer 3 Report
The paper seeks to introduce an approach; Improvement of Mechanical and Tribological Behaviors of GLC Films on NBR under Water Lubrication by Co-doping Ti and N. However, the authors should consider to improve upon the quality to further highlight and emphasis.
1. Could you consider rephrasing the title of the article as “improving the mechanical and tribological behaviors of GLC films on NBR under water lubrication by co-doping of Ti and N.
2. Based on the understanding of what should be included in the abstract, consider adding one or two lines highlighting what the problem you would like to solve.
3. The introduction needs to be improved by relating to the mechanics of the studied materials and their mechanical characteristics. The references to be included are: 10.1007/s10853-022-06994-3, 10.1177/0021998318790093, 10.1016/j.polymertesting.2017.09.009, 10.1177/07316844211051733, 10.1016/j.compstruct.2021.114698, 10.1177/0731684417727143 and 10.1002/app.46770.
4. Abstract of any scientific articles should address three main elements;
i. Description of the problem
ii. Description of the research findings
iii. Significance of the study
In view of these, introduce one or two lines to highlights the significance of your study
5. The footers (magnification bar) in the SEM images are not visible. Manually indicate each inside the figures
6. Figure 4, 5, 7, 9, 10, 12, and 13 were poorly shown. Values and the lines are hard to read. Consider making it more visible
7. There should be a uniformity in the spelling of the word figures. It’s either you abbreviate all or write in full. Consider adopting one style.
Round 2
Reviewer 1 Report
The authors have taken up all the suggestions for improvement and thus significantly increased the quality of the paper.
One thing should be changed: The term "co-doping" in the title is misleading, as one immediately thinks of doping with cobalt. It would be better to write "... by doping with Ti and N". (--> minor revision)
Author Response
Thank you for your comments!They do great help to my paper !
The word co-doping is replaced by doping across the paper!
Have a nice day!
Reviewer 2 Report
Thank you very much for the opportunity to review this paper. The authors have taken my comments into consideration and I believe that the paper in its present form can be accepted for the journal Coatings.
Author Response
Thank you for your professional comments!
Best regards!